# Nonlinear Conductivity and Space Charge Characteristics of SiC/Silicone Rubber Nanocomposites

**DOI:** 10.3390/polym14132726

**Published:** 2022-07-03

**Authors:** Ming-Ze Gao, Zhong-Yuan Li, Wei-Feng Sun

**Affiliations:** 1Department of Electrical Engineering and Electronics, School of Medical Imaging, Mudanjiang Medical University, Mudanjiang 157011, China; gaomingze00@163.com; 2Electric Power Research Institute, State Grid Heilongjiang Electric Power Co., Ltd., Harbin 150080, China; 3School of Electrical and Electronic Engineering, Nanyang Technological University, Singapore 639798, Singapore; weifeng.sun@ntu.edu.sg

**Keywords:** cable joint, liquid silicone rubber, nonlinear conductivity, space charge

## Abstract

To achieve a preferable compatibility between liquid silicone rubber (LSR) and cable main insulation in a cable accessory, we developed SiC/LSR nanocomposites with a significantly higher conductivity nonlinearity than pure LSR, whilst representing a notable improvement in space charge characteristics. Space charge distributions in polarization/depolarization processes and surface potentials of SiC/LSR composites are analyzed to elucidate the percolation conductance and charge trapping mechanisms accounting for nonlinear conductivity and space charge suppression. It is verified that SiC/LSR composites with SiC content higher than 10 wt% represent an evident nonlinearity of electric conductivity as a function of the electric field strength. Space charge accumulations can be inhibited by filling SiC nanoparticles into LSR, as illustrated in both dielectric polarization and depolarization processes. Energy level and density of shallow traps increase significantly with SiC content, which accounts for expediting carrier hopping transport and surface charge decay. Finite-element multiphysics simulations demonstrate that nonlinear conductivity acquired by 20 wt% SiC/LSR nanocomposite could efficiently homogenize an electric field distributed in high-voltage direct current (HVDC) cable joints. Nonlinear conductivities and space charge characteristics of SiC/LSR composites discussed in this paper suggest a feasible modification strategy to improve insulation performances of direct current (DC) cable accessories.

## 1. Introduction

Recent electrical energy supply raises the requirements for delivering electricity over long distances in which traditional high-voltage alternating current (AC) transmission systems are facing inevitable challenges. Compared with conventional AC transmission, the major advantages of high-voltage direct current (HVDC) transmissions, such as long transmission distances, low losses, and large transmission capacities, need to competently provide long-distance transmissions [1,2,3,4].

HVDC cable accessories are essential connecting devices in power transmission and transformation systems. However, the multi-layer structures of composite insulation in cable accessories lead to great discrepancies and incompatibilities in the electrical conductivities of internal insulation materials. In particular, the conductivity of accessory insulation is far lower than that of the main insulation cross-linked polyethylene (XLPE), where the electric field is distributed inversely proportional to the material conductivity [5,6]. As a result, the poor accessory insulation results in local electric field distortions and space charge accumulations under long-term polarization electric fields, which degrades the insulation and breaks down the resistance of cable accessories. Therefore, cable accessories are the weakest insulation points in direct current (DC) cable lines.

Recently, homogenizing the electric field by optimizing the geometry of the insulation structure and using nonlinear-conductivity composite materials has been found capable of alleviating local electric field concentration, improving the insulation performances of cable accessories [7,8]. With the development of large-capacity and high-voltage power equipment, nonlinear materials have shown significant advantages for improving practical efficiency and ensuring electrical compatibility in insulation equipment. The SiC/silicone rubber (SiR) composites with SiC content greater than 20 vol% exhibit evident conductivity nonlinearity which is attributed to carrier hopping conductance introduced by SiC fillers [9]. The dielectric properties of ZnO/SIR composites show significant conductivity nonlinearity when the volume fraction of spherical ZnO fillers is higher than 30 vol%, with relative permittivity being increased by five times [10]. It has been reported that calcium copper titanate (CaCu_3_Ti_4_O_1_) nanofibers/liquid silicone rubber composites present notable nonlinear conductivity and an improvement of space charge characteristics due to the introduced charge traps by CaCu_3_Ti_4_O_1_ nanofillers [11]. SiC nanofillers with nonlinear conductivity can effectively suppress interface charge accumulations in SiC/polymer nanocomposites by introducing shallower traps [12]. In addition, coatings prepared by filling inorganic nonlinear nanoparticles (e.g., SiC and ZnO) into polymers exhibit self-adaptive conductivity in response to the applied electric field. The coating of SiC/Epoxy resin on insulator surfaces could promote the dissipation of surface charge and effectively improve the flashover voltage [13].

The previous studies concerning electric field homogenization using nonlinear additive fillers mainly focused on the influence of the type and content of fillers on the nonlinear conductivity of internal insulation, without elucidating clear pictures of nonlinear conductance in combination with space charge characteristics. Therefore, in this study, SiC nanoparticles filled into liquid silicone rubber (LSR) were used to render nanocomposites with appreciable nonlinear conductivities. The abnormally varying carrier transport and space charge suppression are analyzed to reveal the underlying mechanism of the comprehensive improvements in both nonlinear conductance and insulation strength, which provides a basic reference for the engineering application of nonlinear composites.

## 2. Materials and Methods

### 2.1. Material Preparation

Two-component (A and B) liquid silicone rubber (LSR, Wacker, Germany) was selected as the raw material for the composite matrix. SiC material (Deke Daojin, Beijing, China) with a particle size of 40 nm was adopted as the filling additive. SiC/LSR nanocomposites with SiC contents of 0, 5.0, 10.0, and 20.0 wt% were prepared by the mechanical stirring method. The preparation process was as follows: (1) same masses of A and B, and appropriate mass of SiC nanoparticles were accurately weighed according to filling content to be poured into an beaker and stirred continuously at room temperature; (2) the SiC/LSR mixture was poured into a 10 mm × 10 mm × 0.2 mold in a vacuum environment at a constant room temperature for 2 h; and (3) the mold containing the SiC/LSR mixture was placed into a flat vulcanizing instrument and vulcanized for 15 min under 15 MPa at 393 K, and then cured at 473 K for 4 h to finally obtain the material samples.

### 2.2. Microstructure Characterization

The insulation properties of nanodielectrics are closely related to the dispersion of the doped nano-phases in a polymer matrix. Thus, the dispersion of the SiC nanofillers in an EPDM polymer matrix was characterized by a scanning electron microscope (SEM) (SU8020, Hitachi High-Technologies Corporation, Tokyo, Japan). The specimens were cold-brittle ruptured in a vessel containing liquid nitrogen to acquire the cross section to be observed by SEM after being sprayed with a gold film.

### 2.3. Mechanical Tensile Test

According to the standard of ISO 37:2005, the stress–strain characteristics were measured with an elongation speed of 5 mm/min. The tested sample was made into dumbbell type sample 1 mm thick, with a total length of 75 mm and an effective test width of 4 ± 0.2 mm. The surface was smooth. without visible defects.

### 2.4. Electric Conductivity

Electric conductivities of the SiC/LSR nanocomposites in samples of 50 mm diameter and 300 µm thickness were measured with a three-electrode system consisting of a high-voltage DC power supply (HB-Z103-2AC, Tianjin Hengbo High Voltage Power Supply Co., Ltd., Tianjin, China), a picoammeter (Est122), three electrodes, and an oven at temperatures from 30 to 70 °C. The current values were predicted approach a quasi-steady sate after applying voltage for 30 min. Three aluminum electrodes were evaporated in a vacuum in which the measuring disc-shaped electrode with a 50 mm diameter was encircled by a protective annular-shaped electrode with a 75 mm diameter on one side of the film samples; a high-voltage circular electrode with a diameter of 78 mm was placed on the other side. Each group of samples was measured several times, and the average value was taken to ensure the accuracy and reliability of the tested results.

### 2.5. Space Charge Distribution

The pulsed electro-acoustic (PEA) method was used to measure the space charge characteristics of the SiC/LSR composites in samples with a dimension of 50 mm × 50 mm × 0.3 mm at room temperature in which the polarization electric field, the pulse voltage and width, and the input impedance were specified as 30 kV/mm, 400 V and 8 ns, and 1.368 Ω, respectively. After voltage was applied for 30 min to test the space charge accumulation in the polarization process, the tested sample was short-circuited to obtain depolarization decaying of the space charge distribution. The total space charge *Q*(*t*) was calculated as follows [14]:(1)$$Q(t)=\int_{0}^{h}\left|\rho (x,t)\right|Sdx$$
where *Q*(*t*) denotes the total charge quantity internal material at time *t*, *h* signifies the sample thickness, *ρ*(*x*,*t*) symbolizes the space charge density at the position of *x* in the sample at *t*, and *S* is the electrode area.

Considering the small space charge injection under a low electric field, the waveform at a low electric field intensity of 3 kV/mm was adopted as the waveform reference for the space charge measurement system to restore the measured signal of a space charge distribution under a high electric field.

### 2.6. Isothermal Surface Potential Decay

The isothermal surface potential decay (ISPD) method can characterize the energy level distribution of charge traps and the charge transport in a detrapping process in which the corona-discharge method is used to charge the film sample, as shown by the schematic measurement system in Figure 1.

The distance between the needle tip and the grid was 5 mm, and the distance between the grid and the sample surface was 5 mm. The needle and grid electrodes were connected to a high-voltage source with an aluminum back electrode being grounded during measurement. The electrostatic probe was fixed on the sliding guide bracket about 3 mm from the sample surface. Under negative corona charging, the needle and grid electrodes were charged by −8 kV and −5 kV voltage, respectively, for 5 min. Then, the sample was transferred below an electrostatic probe to be measured for 25 min.

Double exponential function was used to numerically fit the decaying surface potential over time very well, as expressed by:(2)$$U=Ae^{\frac{-t}{\tau_{1}}}+Be^{\frac{-t}{\tau_{2}}}$$
where A, *τ*_1_, B, and *τ*_2_ are fitting parameters by which the trap energy level distribution can be calculated by the surface potential decaying curve [14,15] which could be considered as the volume trap characteristics. The trap energy level and trap density are calculated by the following formulas:(3)$$E_{t}=k_{b}T\ln(\nu t),\;N(E)=\frac{4\varepsilon_{0}\varepsilon_{1}}{qk_{b}Td^{2}}\left|t\frac{dU}{dt}\right|$$
where *E_t_* denotes trap energy level, *N*(*E*) indicate trap density, *d* represents sample thickness, *ε*_0_ and *ε*_1_ are vacuum permittivity and relative permittivity, respectively, *v* symbolizes the electron escape frequency, and *q* and *k*_b_ signify the electron charge and Boltzmann constant, respectively.

### 2.7. Finite-Element Electric Field Simulation

We investigates the electric field distributions in the 200 kV DC cable joint with the SiC/LSR composites used as reinforced insulation, as schematically shown by the geometrically modeled structure in Figure 2. The diameter of the cable core was 30 mm, and the thicknesses of the main insulation (XLPE material), inner shield, outer shield, and reinforced insulation were specified as 16, 2, 1, and 68 mm, respectively. The length of the simulation model was 2600 mm. To make the simulation more identical to the actual situation, an air environment domain was added to the simulation model. The convective heat transfer parameter at the interface of the cable joint and the air was set to 10 W/(m^2^·K). DC high voltage was loaded to the edge of the cable core. The inner edge of the outer semi-conductive shield layer of insulation and the boundary of the stress–cone were set as the electric ground boundary. For thermal simulations, which were coupled to the electrical field based on the electrical properties of each conduction or insulation component, the core and ambient (outside outer protective layer) temperatures were set as 70 °C and 30 °C, respectively. The thermal–electrical coupling simulations were implemented by COMSOL Multiphysics according to reference [16], employing material parameters of electrical and thermal parameters for each constituent as listed in Table 1.

Numerical simulations with the finite-element method were performed to calculate the electric field distributed in the cable accessory using the modified electrical properties of reinforced insulation materials characterized by the prepared SiC/LSR nanocomposites, such as the nonlinear conductivity and the higher relative permittivity than cable main insulation materials as listed in Table 1. A free triangular cell was utilized for finite-element meshing by the Delaunay triangulation algorithm, which was refined locally at the positions where the electric field strength varies significantly in the cable terminal. The maximum and minimum numbers of elements were adjusted until no obtuse angles in the triangulation meshing process appeared. The element growth rate was specified as 1.5, which means that the element size increased by about 50% from one element to another. The slack in narrow regions was set as 1.0 to prevent the triangular meshing from generating different sizes of elements.

## 3. Results and Discussion

### 3.1. Microstructure and Mechanical Property

The microscopic morphologies of the SiC/LSR nanocomposites observed by SEM are shown in Figure 3 in which the highlighted areas indicate SiC nanofillers. For 5.0 wt% filling content, SiC nanofillers are evenly dispersed without any obvious agglomeration. With the increase of SiC content up to 20.0 wt% content, the highlighted parts inside the nanocomposites increase slightly in size, while retaining a high dispersivity of homogeneous spatial distributions, as shown in Figure 3. The SEM images verify that the SiC/LSR nanocomposites were successfully prepared as expected to maintain a high dispersion of SiC nanofillers even for the filling content approaching 20.0 wt%.

The tensile modulus and broken-elongation of the SiC/LSR nanocomposites, as illustrated by the stress–strain characteristics in Figure 4, increase and decrease, respectively, with the increase of SiC content. When SiC content approached 20.0 wt%, the tensile strength and broken-elongation of SiC/LSR nanocomposites were maintained at 6 MPa and 350%, respectively, which meet the mechanical requirement for cable accessories. However, due to the fact that the mechanical performances of cable accessories are greatly influenced by operating conditions and cable manufacturing technology, the applications of the SiC/LSR nanocomposites need to be further investigated for improving mechanical properties.

### 3.2. Electrical Conductance

Electrical conductivity was normalized as the dependence of the current density on the electric field (*J*-*E* variation curves) to investigate the charge transport characteristics of the SiC/LSR nanocomposites, as shown by the *J*-*E* curves of the SiC/LSR composites in Figure 5. The critical points arise around electric field strength of 10^7^ V/m, as a threshold *E*_th_ at which point the charge transport mechanism changes from Ohmic conductance to space charge limited conductance (SCLC). Nonlinear coefficients *β*_1_ and *β*_2_ distinguished by *E*_th_ are obtained by the linear fitting for the logarithm *J*-*E* curves, as listed in Table 2.

When *E* < *E*_th_, the nonlinear coefficient *β*_1_ < 2 complies with the Ohmic conductance mechanism that the charge transport is produced by impurity ionization in the SiC/LSR composites. In contrast, when *E* > *E*_th_, the nonlinear coefficient *β*_2_ > 2 which means the carrier concentration and carrier mobility increase drastically. The process of carrier increment in polymers is a thermal excitation process. According to hopping conductance theory, most of the charges involved in conducting current are in local states, and the charge transfer between localized states is the main process. The process of charge passing from one localized state to another is graphically described as hopping conductance, the current density formula of the jump conductance model can be described as follows:(4)$$J_{n}=2nd\upsilon e\exp\left(-\frac{\chi }{k_{b}T}\right)\sinh\left(\frac{elE}{2k_{b}T}\right)$$
where *n* represents carrier concentration, *χ* denotes activation energy, *l* symbolizes hopping distance of the carrier, and *T* is thermodynamic temperature. To judge whether the conduction mechanism under a high electric field is dominated by hopping transport, the *J*-*E* curves of Figure 5a,b are fitted by Equation (4) according to the hopping conductance model, as shown in Figure 5c,d. The hopping distance of the SiC/LSR composites can be seen in Table 3. The fitting curves are highly consistent with the tested conductivities of the SiC/LSR nanocomposites. Based on the fitting curves in the hopping conductance model, the hopping distance of 5.0 wt%, 10.0 wt%, and 20.0 wt% for the SiC/LSR nanocomposites are calculated by Equation (4), as shown in Table 3. The hopping distance becomes evidently larger with the increase in SiC content, implying that SiC nanofillers facilitate the charge detrapping and thus expedite hopping transports to acquire or increase conductivity nonlinearity. Therefore, the higher the temperature, the higher the density of trap that can be disentangled, and the shorter the hopping distance. Moreover, according to the classical percolation theory [17], when the content of SiC nanoparticles in the LSR matrix exceeds the percolation threshold, the SiC nanofillers are close enough to form a random conductive network, resulting in the macroscopic performance of nonlinear conductivity of the SiC/LSR nanocomposites.

### 3.3. Space Charge Characteristics

In high-voltage DC cable systems, the XLPE cable and its cable accessories bear a high electric field for long periods in which the insulation layer suffers space charge accumulation under electric field distortion at the interface of the multilayered composite insulation structure, accelerating dielectric aging and finally causing insulation failures. For cable accessories, the maximum electric field intensity resides at the stress-cone root of reinforced insulation. This is the part that requires high resistance to space charge accumulations. Space charge characteristics of pure LSR and SiC/LSR nanocomposites in polarization and depolarization processes are shown in Figure 6. Space charge accumulation appears inside the pure LSR sample in the polarization process, which can be significantly inhibited by filling SiC nanoparticles and becomes more obvious for higher SiC content, as comparatively illustrated by the left panels of Figure 6a. Due to the lower mass of negative carriers compared to positive carriers, it is more likely that the negative charge can approach the anode without neutralization in the electrical migration process, accounting for the evident heterocharge accumulations arising near the anode which can be inhibited by filling SiC particles and alleviated with the increase of SiC content.

Homocharge and heterocharge as two classes of space charges are accumulated by trapping the electrode-injected carriers and the impurity ionized carriers that form the applied electrode and the dielectric material interior, respectively [18,19,20]. Therefore, in the depolarization process, heterocharges decay more quickly than homocharges under a short-circuit as shown in the right panels of Figure 6. There, the space charge decaying near the anode is rather faster than that of the homocharges near the cathode. Meanwhile, the charge density of pure LSR at the initial stage of short-circuit depolarization is much higher than that of the SiC/LSR nanocomposites in which charge density has been further decreased by the increasing SiC content.

Mean space charge densities in pure LSR and SiC/LSR nanocomposites during a depolarization process are calculated by Equation (1), with the results being shown in Figure 7. The space charges decaying with an almost constant rate in the initial depolarization stage of 0~300 s derives mainly from the detrapping of the charges captured in shallow traps, while the significantly lower decaying rate of the space charge density after depolarization for 300 s implies that the space charge decaying is derived from detrapping the charges captured in deeper traps. In particular, it is evident that the space charges density of the SiC/LSR nanocomposites decreases with the increase in SiC content. In comparison with pure LSR, SiC/LSR nanocomposites acquire a higher resistance to space charge accumulation, while persisting at a higher depolarization rate due to the higher electrical conductance derived from carrier hopping transport through the conductive channels formed between SiC nanofillers.

### 3.4. Charge Trap Characteristics

Surface charge dissipation is generally fulfilled by transporting along the material surface or from interior material toward electrode neutralization with charged particles in gas material. As indicated by Figure 8a, the SiC/LSR nanocomposites have achieved a notably higher rate of surface charge dissipation than pure LSR. For the electrode structure and the sample size used for our study, the strength of the normal electric field between the sample surface and the ground electrode is much higher than that of tangential electric fields along the sample and electrode surfaces in the process of surface charge dissipation. This process is dominated by the surface charge transports through the trapping and detrapping processes in the sample interior (volume conduction) to the ground electrode. The potential decay rate of the material surface increases significantly with the increase in SiC content due to the hopping transport of the charges percolated from the traps introduced by the SiC nanofillers. This is manifested by the higher threshold electric field of the SiC/LSR nanocomposites compared with pure LSR at which the electric conduction alters into the highly nonlinear region. Space charge characteristics or surface charge dissipation of the SiC/LSR nanocomposites is dominated by carrier trapping and de-trapping processes. Meanwhile, the energy level and spatial density of trap distributions determine the activation energy (carrier concentration) and hopping distance (carrier mobility) of percolation conductance in the nonlinear region. According to ISPD curves, we calculate trap level distributions of pure LSR and the SiC/LSR nanocomposites by Equations (2) and (3), as shown by the results in Figure 8b.

At about 0.86 and 0.93 eV of trapping depth, two peaks of trap density appear in the energetic spectra for the SiC/LSR nanocomposites, which could be referenced as the shallow and deep trap levels, respectively. It is noted that the SiC/LSR nanocomposites present distinctly lower and considerably higher densities of deep and shallow traps, respectively, compared with pure LSR, which is more obvious for higher SiC content. The multi-core model [21,22], which is competent to characterize the interface region between the nanofillers and the polymer matrix in nanodielectrics, is composed of a bonding layer, a binding layer, and a loose layer. In the bonding and binding layers, deep traps are derived from the imperfect chemical bonding in the molecular structure, while the amorphous structure in the loose layer mainly contributes to shallow traps. When the filling content is raised to a sufficient level, generally higher than 5.0 wt%, the interface areas as described by the loose layer between the SiC nanofillers overlap to form conductive channels for hopping transports of the charge carriers detrapped from shallow traps, as described by the percolation effect. The significant increase of shallow trap density by filling SiC nanoparticles results in the substantial percolation conductance at room temperature, which will be aggravated by increasing temperature or SiC content, accounting for the extraordinary conductivity nonlinearity given by the SiC/LSR nanocomposites.

### 3.5. Electric Field Simulation

For DC cable in operation, the temperature in the metallic cable core would be raised by Joule heating, leading to a specific temperature gradient from the cable core to the external insulation. The conductivity of polymer insulation materials, which determines the electric field distribution of the DC cable joint, depends greatly on operation temperature and heat production. For pure LSR or SiC/LSR composites as the reinforced insulation material, the electric field distributions in a 200 kV DC cable joint are simulated under a thermal–electrical fully coupling condition, as shown in Figure 9.

Electric field distribution is inversely proportional to conductivity under DC voltage, and the conductivity of pure LSR is much lower than that of XLPE. Thus, high electric field is distributed in LSR reinforced insulation, as shown in Figure 9a. Maximum electric field strength appearing at the stress-cone root approach 43.40 kV/mm when pure LSR is used for reinforced insulation, which will be effectively reduced by using SiC/LSR nanocomposites as reinforced insulation material, as shown in Figure 9b,c. The position of maximum electric field strength can be ameliorated by locating it in the main XLPE insulation when using the SiC/LSR nanocomposite with a high SiC content approaching 20 wt%, as shown in Figure 9d.

## 4. Conclusions

The effect of SiC nano fillers on the conductivity and the charge trap characteristics of LSR composites was investigated in this study. Conductivity nonlinearity and threshold electric field strength of SiC/LSR nanocomposites increases and decreases, respectively, with the increase of SiC content, which will be intensified at a higher temperature. The highest density peak of trap level distribution shifts from deeper traps to shallow traps by introducing SiC nanofillers into the LSR matrix, which accounts for hopping conductance and impeding space charge injection. Based on the established cable joint model of finite-element multiphysics simulations, it is concluded that the highest electric field strength at the stress-cone root can be effectively reduced by using SiC/LSR nanocomposites as reinforced insulation, compared with using pure LSR material, which can even be intensified by raising the SiC content. As a consequence, the approximate SiC particles doped LSR can suppress the accumulation of space charge and improve conductivity nonlinearity, which provides a possible method for the improvement of cable accessory performance.

## Figures and Tables

**Figure 1 polymers-14-02726-f001:**
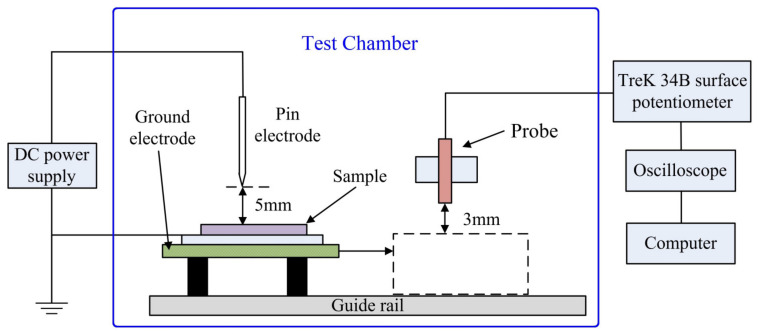
Schematic test system of isothermal surface potential decay.

**Figure 2 polymers-14-02726-f002:**
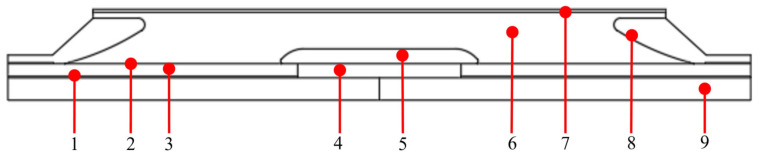
Geometry model of cable joint: 1, inner shield; 2, outer shield; 3, XLPE; 4, connection fittings; 5-high-voltage shielding tube; 6, reinforced insulation; 7, outer protective layer; 8, stress cone; 9, conductor core.

**Figure 3 polymers-14-02726-f003:**
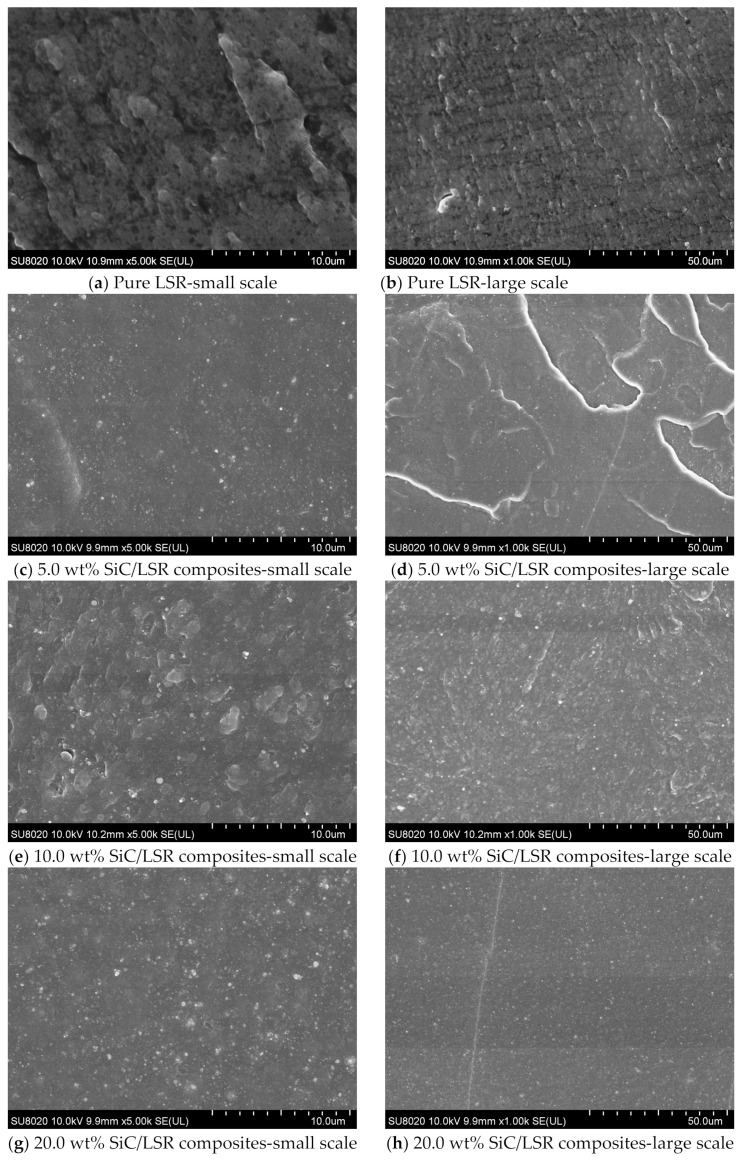
SEM images of SiC/LSR nanocomposites.

**Figure 4 polymers-14-02726-f004:**
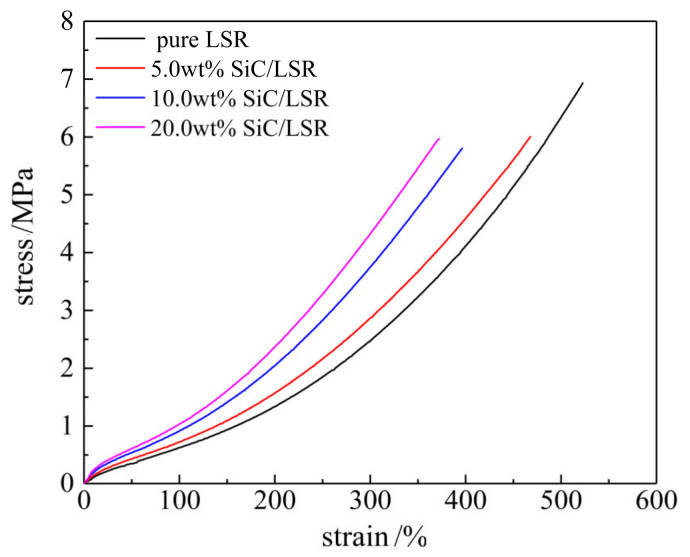
Stress–strain characteristics of SiC/LSR nanocomposites.

**Figure 5 polymers-14-02726-f005:**
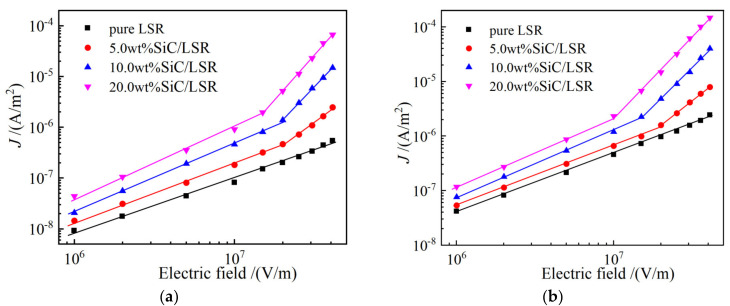
Electric conductance characteristics in the logarithm *J*-*E* curves of pure LSR and SiC/LSR nanocomposites at diverse temperatures of: (**a**) 30 °C; and (**b**) 70 °C; hopping conductance characteristics of fitting conductivities for the SiC/LSR nanocomposites at (**c**) 30 °C; and (**d**) 70 °C.

**Figure 6 polymers-14-02726-f006:**
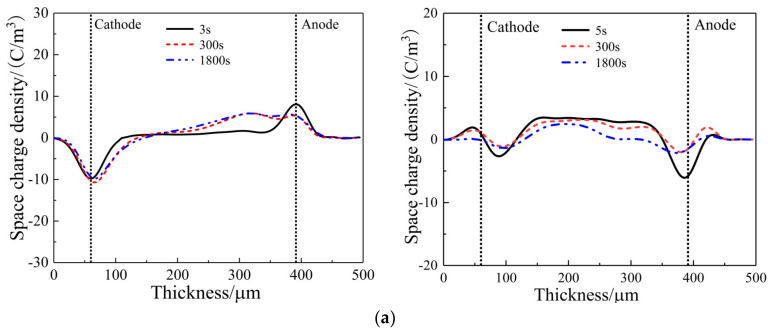
Space charge distribution in: (**a**) pure LSR; (**b**) 5 wt%; (**c**) 10 wt%; and (**d**) 20 wt% SiC contents in the polarization process of applying voltage (left panels) and in the depolarization process under a short-circuit (right panels).

**Figure 7 polymers-14-02726-f007:**
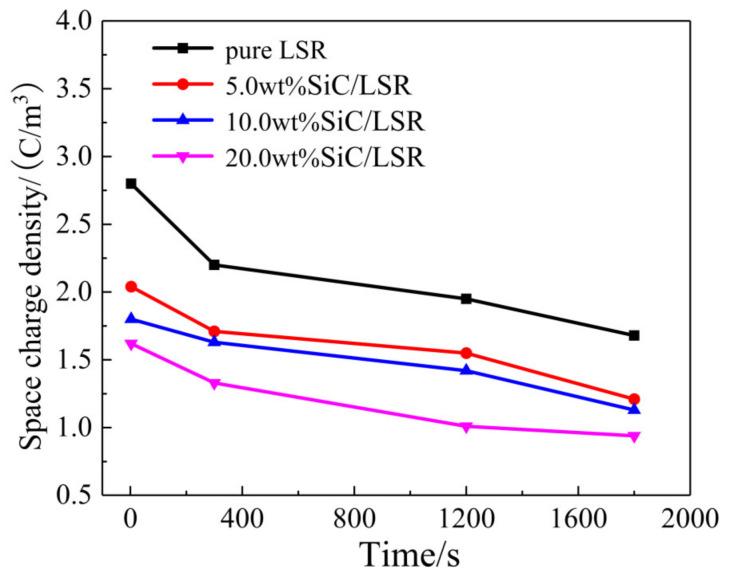
Space charge density of pure LSR and SiC/LSR nanocomposites during depolarization.

**Figure 8 polymers-14-02726-f008:**
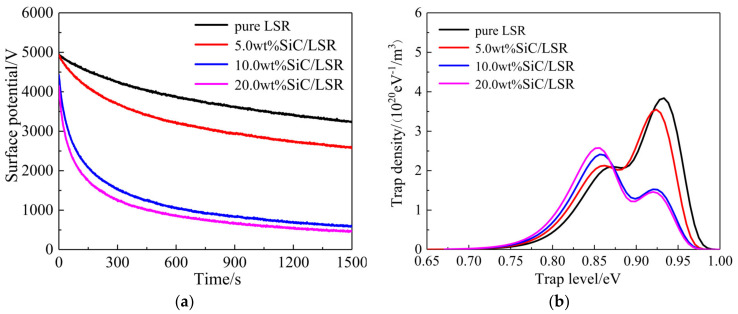
(**a**) Isothermal surface potential decay (ISPD) curves; and (**b**) trap level distributions of SiC/LSR nanocomposites.

**Figure 9 polymers-14-02726-f009:**
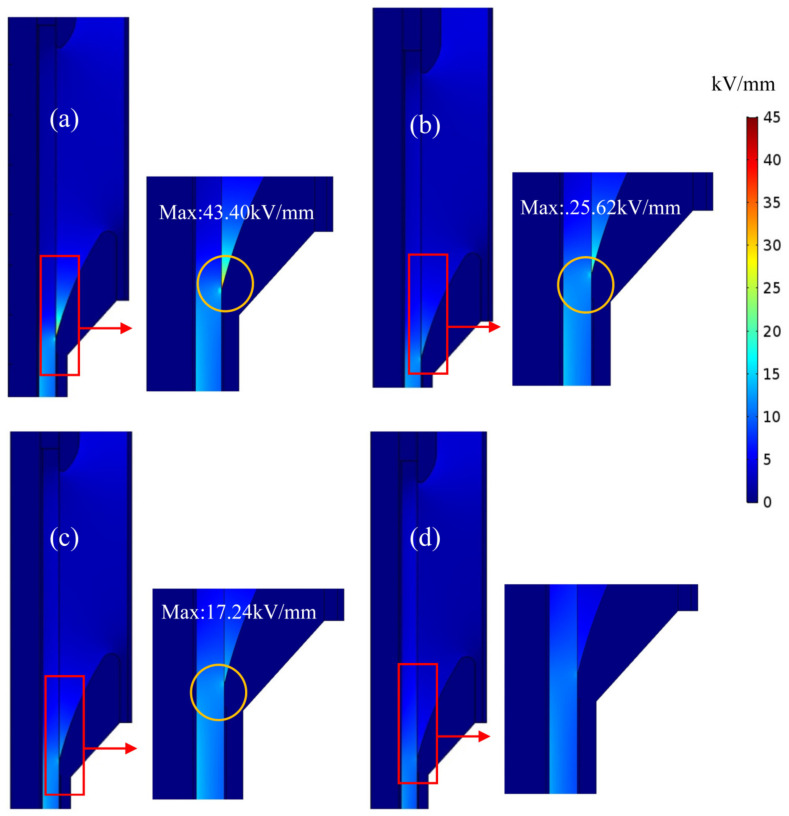
Steady-state electric field distributions in a cable joint with reinforce insulation of: (**a**) pure LSR and SiC/LSR nanocomposites of (**b**) 5 wt%; (**c**) 10 wt%; and (**d**) 20 wt% SiC contents.

**Table 1 polymers-14-02726-t001:** Electrical and thermal parameters of materials specified in the electric field simulations.

Materials	Density/(g·cm^−3^)	Relative Permittivity	Coefficient of Thermal Conductivity/(W·m^−^^1^·K^−^^1^)
XLPE	910	2.27	1640
Inner Shield	950	100	2500

**Table 2 polymers-14-02726-t002:** Nonlinear coefficients of the SiC/LSR nanocomposites.

Materials	30 °C		70 °C	
	*E*_th_ (kV/mm)	*β*_1_, *β*_2_	*E*_th_ /(kV/mm)	*β*_1_, *β*_2_
pure LSR	-	-	-	-
5.0 wt%	21.1	1.14, 2.31	10.1	1.10, 2.26
10.0 wt%	20.4	1.37, 3.31	14.9	1.23, 2.85
20.0 wt%	14.8	1.37, 3.56	10.5	1.29, 2.99

**Table 3 polymers-14-02726-t003:** Hopping distance of percolation conductance in the SiC/LSR composites.

Materials	30 °C	70 °C
5.0 wt% SiC/LSR	4.0	3.5
10.0 wt% SiC/LSR	5.0	4.9
20.0 wt% SiC/LSR	5.4	5.2

## Data Availability

Theoretical methods and results are available from the authors.

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
