# Peer review of "Nonlinear Conductivity and Space Charge Characteristics of SiC/Silicone Rubber Nanocomposites"

_polymers, 2022, doi:10.3390/polym14132726_

Round 1
Reviewer 1 Report
This research is about the sic/silicone rubber nanocomposites regarding the conductivity and space charge features.
1. it is nice to see the non-linear trend regarding electrical conductivity - however, for practical applications, mechanical robustness is important. Thus, the author should provide the mechanical test data, such as the tension or compression test results, or the dynamic mechanical analysis.
2. The electrical field simulation is straightforward - can the author list the reference for the coefficient of thermal conductivity in table 1? or provide the test method if this is from the author's test.
Author Response
(to Reviewer 1)
- it is nice to see the non-linear trend regarding electrical conductivity - however, for practical applications, mechanical robustness is important. Thus, the author should provide the mechanical test data, such as the tension or compression test results, or the dynamic mechanical analysis.
Revise: Stress-strain characteristics are complemented as indicated by the Section 2.3 and 3.1 in the revised manuscript :
2.3. Mechanical Tensile Test
According to the standard of ISO 37:2005, the stress-strain characteristics are measured with a elongation speed of 5mm/min. The tested sample is made into dumbbell type sample 1 mm thick, with a total length of 75 mm and an effective test width of 4±0.2 mm. The surface is smooth without visible defects.
3.1 Microstructure and mechanical property
---
Tensile modulus and broken-elongation of SiC/LSR nanocomposites, as illustrated by stress-strain characteristics in Figure 4, increases and decreases respectively with the increase of SiC content. When SiC content approach 20.0wt%, the tensile strength and broken-elongation of SiC/LSR nanocomposites maintain to be 6MPa and 350% respectively, which meet the mechanical requirement for cable accessories. However, due to the mechanical performances of cable accessories are greatly influenced by operation condition and cable manufacturing technology, the applications of SiC/LSR nanocomposites needs to further investigate for improving mechanical properties.
Figure 4. Stress-strain characteristics of SiC/LSR nanocomposites
- The electrical field simulation is straightforward - can the author list the reference for the coefficient of thermal conductivity in table 1? or provide the test method if this is from the author's test.
Revise: As indicated at the end of 1st paragraph in Section 2.7 of revised manuscript, the reference [16] (Li, C.M.; Wu, G.F.; Li, C.Y. Effect of the defects inside XLPE insulated HVDC cable termination on the electric field distribution. Electric Machines and Control 2018, 12(22), 63-67) is cited as “According to reference [16], employing material parameters of electrical and thermal parameters for each constituent as listed in Table 1.”

Reviewer 2 Report
This paper presents an analysis of the properties of SiC / LSR nanocomposites used in cable insulation systems.
The paper may be of some interest to the scientific community;
- The Introduction section needs to be substantially improved as no in-depth analysis of the results of other research in this field is being carried out at this stage. This section focuses on the presentation of the properties of certain types of materials, which is not enough.
- In the Materials section, a detailed characterization of the properties of the materials used and the presentation of macroscopic images for the test pieces made must be made.
- Research methodology needs to be improved as it is not enough to present that 5% nanocomposites have been made; 10%; 20% SiC. It is necessary to establish an experimental research program of DOE type;
-In the case of Figure 3, it would not be understood why in cases c and d the LSR was not taken into account.
-FEM analysis must be substantially detailed. It is necessary to present the systems of restrictive conditions used but also the software used;
- A microscopic analysis of the structure of the tested samples is also required;
- The section on experimental research is not sufficiently presented, and the processing of experimental data needs to be further developed. It is not enough to consider only those types of samples because the results are not relevant under these conditions;
- The discussion part must be much developed in order to be able to highlight the novelty brought by the research presented in the paper in relation to other research in the field. In this form in this section no report can be identified on the results obtained compared to previous research;
- Conclusions should be more concrete and present future research directions.
Author Response
(to Reviewer 2)
- 1. The Introduction section needs to be substantially improved as no in-depth analysis of the results of other research in this field is being carried out at this stage. This section focuses on the presentation of the properties of certain types of materials, which is not enough.
Revise: Accordingly, three statements and cited references are incorporated in 3rd paragraph of Introduction as “It has been reported that calcium copper titanate (CaCu3Ti4O1) nanofibers/liquid silicone rubber composites represent a notable non-linear conductivity and an improvement of space charge characteristics due to the introduced charge traps by CaCu3Ti4O1 nanofillers [11]. SiC nanofillers with nonlinear conductivity can effectively suppress interface charge accumulations in SiC/polymer nanocomposites by introducing shallower traps [12]. In addition, coatings prepared by filling inorganic non-linear nanoparticles (e.g. SiC and ZnO) into polymers exhibit self-adaptive conductivity in response to the applied electric field. The coating of SiC/Epoxy resin on insulator surface could promote the dissipation of surface charge and effectively improve the flasover voltage [13].”
- In the Materials section, a detailed characterization of the properties of the materials used and the presentation of macroscopic images for the test pieces made must be made.
Revise: SEM characterizations and its method are complemented as indicated by the Section 2.2 and 3.1 in the revised manuscript as:
2.2. Microstructure characterization
The insulation properties of nanodielectrics are quite related to the dispersion of the doped nano-phases in polymer matrix. Thus, the dispersion of SiC nanofillers in EPDM polymer matrix is characterized by a scanning electron microscope (SEM) (SU8020, Hitachi High-Technologies Corporation, Tokyo, Japan). The specimens are cold-brittle ruptured in a vessel containing liquid nitrogen to acquire the cross-section to be observed by SEM after being sprayed with a gold film.
3.1 Microstructure and mechanical property
The microscopic morphologies of SiC/LSR nanocomposites observed by SEM are shown in Figure 3, in which the highlighted areas indicate SiC nanofillers. For 5.0wt% filling content, SiC nanofillers are evenly dispersed without any obvious agglomeration. With the increase of SiC content up to 20.0wt% content, the highlighted parts inside nanocomposites increase slightly in size, while retaining a high dispersivity of homogeneous spacial distributions, as shown in Figure 3. The SEM images verify that SiC/LSR nanocomposites have been successfully prepared as expected to maintain a high dispersion of SiC nanofillers even for the filling content approaching 20.0wt%.
|
(a) Pure LSR-small scale |
(b) Pure LSR-large scale |
|
(c) 5.0wt%SiC/LSR composites-small scale |
(d) 5.0wt%SiC/LSR composites-large scale |
|
(e) 10.0wt%SiC/LSR composites-small scale |
(f) 10.0wt%SiC/LSR composites-large scale |
|
(g) 20.0wt%SiC/LSR composites-small scale |
(h) 20.0wt%SiC/LSR composites-large scale |
Figure 3. SEM images of SiC/LSR nanocomposites
Furthermore, because the macroscopic photos of prepared materials show not discrepancy between composites with different contents, we do not present these pictures, while, we supplement the details of material preparation processes as indicated in Section 2.1:
“Two-component (A and B) liquid silicone rubber (LSR, Wacker, Germany) is selected as the raw material for composite matrix. SiC material (Deke Daojin, Beijing, China) with the particle size of 40nm is adopted as the filling additive. SiC/LSR nanocomposites with SiC contents of 0, 5.0, 10.0, and 20.0wt% are prepared by mechanical stirring method. Preparation process is detailed as follows: (1) same masses of A and B, and appropriate mass of SiC nanoparticles are accurately weighed according to filling content to be poured into an beaker and stirred continuously at room temperature; (2) SiC/LSR mixture is poured into 10mm×10mm×0.2 mold in vacuum environment at room temperature persisting for 2 h; (3) the mold containing SiC/LSR mixture is put into a flat vulcanizing instrument and vulcanized for 15min under 15MPa at 393K, and then cured at 473K for 4h to finally obtain the material samples.”
- Research methodology needs to be improved as it is not enough to present that 5% nanocomposites have been made; 10%; 20% SiC. It is necessary to establish an experimental research program of DOE type;The section on experimental research is not sufficiently presented, and the processing of experimental data needs to be further developed. It is not enough to consider only those types of samples because the results are not relevant under these conditions.
Response: According to the results of dc breakdown field strength test, with the increase of SiC content, the nonlinear conductivity of LSR composite becomes more and more obvious, but the DC breakdown field strength of LSR decreases gradually, as indicated by the detailed discussions in the 2nd paragraph of Section 3.2 and the entire Section 3.4. When the SiC content is 20.0wt%, the DC breakdown field strength of LSR composite has decreased to below 50kV/mm, and when the content continues to increase, the breakdown field strength decreases seriously, which is unable to meet demand. In addition, as shown by the complemented Section 3.1, with the increase of SiC content, the highlighted part inside the sample increases significantly, indicating that SiC appears to be agglomerated, but the highlighted part in the figure is still evenly dispersed, if SiC content surpass 20.0wt%, the agglomeration may become more obvious, affecting the properties of LSR composites. So we think, it is not much significant of further developing composites with a filling content higher than 20.0wt%.
- In the case of Figure 3, it would not be understood why in cases c and d the LSR was not taken into account.
Response: This paper mainly considers the mechanism of nonlinear conductance, as indicated by 2nd paragraph of Section 3.2: “When E<Eth, the nonlinear coefficient β1<2 complies with the Ohmic conductance mechanism that the charge transport is produced by impurity ionization in SiC/LSR composites. In contrast, when E>Eth, the nonlinear coefficient β2>2 which means the carrier concentration and carrier mobility increases drastically”. According to the J-E curve of SiC/LSR, the J-E curve of pure LSR has no inflection point without significance of linear fitting, so we choose the conductivity data of 5.0wt%~20.0wt% SiC/LSR nanocomposites
- FEM analysis must be substantially detailed. It is necessary to present the systems of restrictive conditions used but also the software used.
Revise: In Section 2.7, the simulation methodology is supplemented as “The length of simulation model is 2600mm. To make the simulation more identical to the actual situation, air environment domain is added to simulation model. The convective heat transfer parameter at the interface of cable joint and air is set to 10W/(m2·K). DC high voltage is loaded to the edge of cable core. The inner edge of outer semi-conductive shield layer of insulation and the boundary of stress-cone are set as the electric ground boundary.”.
- A microscopic analysis of the structure of the tested samples is also required;
Response and Revise: The SEM characterization analysis has been complemented as indicated by the Section 3.1 in the revised manuscirpt as “The microscopic morphologies of SiC/LSR nanocomposites observed by SEM are shown in Figure 3, in which the highlighted areas indicate SiC nanofillers. For 5.0wt% filling content, SiC nanofillers are evenly dispersed without any obvious agglomeration. With the increase of SiC content up to 20.0wt% content, the highlighted parts inside nanocomposites increase slightly in size, while retaining a high dispersivity of homogeneous spacial distributions, as shown in Figure 3. The SEM images verify that SiC/LSR nanocomposites have been successfully prepared as expected to maintain a high dispersion of SiC nanofillers even for the filling content approaching 20.0wt%.” and the figure of SEM images also be inserted.
- The discussion part must be much developed in order to be able to highlight the novelty brought by the research presented in the paper in relation to other research in the field. In this form in this section no report can be identified on the results obtained compared to previous research.
Response: the novelty of this paper resides in the Section 3.2 and 3.4 focusing on hopping transport in correlation with shallow traps introduced by SiC nanofillers which have been consistently manifested through Section 3.3, and their practical advantages of applying for cable joint are demonstrated by Section 3.4, and have been highlighted in abstract and conclusion. Actually until now, to our knowledge, no pertinent researches have been reported for the mechanism of correlating hopping transport to nonlinear conductance, they all resides in the shallow trapping mechanism without an elucidation as better as the present research.
- Conclusions should be more concrete and present future research directions.
Revise: the conclusion is revised by concrete statements and supplemented with prospectives as “The effect of SiC nano fillers on the conductivity and the charge trap characteristics of LSR composites were investigated in this study. Conductivity nonlinearity and threshold electric field strength of SiC/LSR nanocomposites increases and decreases respectively with the increase of SiC content, which will be intensified at a higher temperature. The highest density peak of trap level distribution shifts from deeper traps to shallow traps due to introducing SiC nanofillers into LSR matrix, which accounts for hopping conductance and impeding space charge injection. Based on the established cable joint model of finite-element multiphysics simulations, it is concluded that the highest electric field strength at stress-cone root can be effectively reduced by using SiC/LSR nanocomposites as reinforced insulation, compared with using pure LSR material, which can be even intensified by raising SiC content. As a consequence, the approximate SiC particles doped LSR can suppress the accumulation of space charge and improve conductivity nonlinearity, which provides a possible method for the improvement of cable accessory performance.”

Round 2
Reviewer 1 Report
accept
Reviewer 2 Report
The authors revised their manuscript according to my suggestions. Thus the manuscript can be accepted for publication.